# Anthropometric and Physiological Profile of Mixed Martial Art Athletes: A Brief Review

**DOI:** 10.3390/sports7060146

**Published:** 2019-06-14

**Authors:** Charalampos Spanias, Pantelis T. Nikolaidis, Thomas Rosemann, Beat Knechtle

**Affiliations:** 1UFR STAPS, University of Nantes, 44322 Nantes, France; spaniasharris@gmail.com; 2Exercise Physiology Laboratory, 18450 Nikaia, Greece; pademil@hotmail.com; 3Institute of Primary Care, University of Zurich, 8006 Zurich, Switzerland; thomas.rosemann@usz.ch

**Keywords:** mixed martial arts, physiology, anthropometrics, exercise testing, muscle power

## Abstract

The aim of this review was to analyze the existing knowledge of the anthropometric and physiological characteristics of MMA (mixed martial arts). A computerized search was performed in the PubMed and Google Scholar databases for English-language, peer-reviewed investigations using the term “mixed martial arts” or “MMA” alone and in combination with “physiological characteristics”, “physiological profile”, “body composition”, “strength”, “power”, “endurance”, “aerobic”, and “anaerobic”. The main findings of this review suggested an overall profile of low body fat, high flexibility, muscle strength, muscle endurance, and anaerobic power, and average cardiovascular endurance. Nevertheless, no differences in anthropometric and physiological characteristics by performance level of the athletes were observed. Based on the studies analyzed in this review, several limitations were reported. First, the subjects in each paper were limited in number, as is MMA literature itself, and it is impossible to make a generalization regarding the anthropometric and physiological profile for male athletes. Second, no studies included female MMA athletes; therefore, there was no evidence for what their profiles might be. Last, the majority of the above-mentioned studies used different tests, or the same tests with different protocols, and as a result, it is difficult to compare their results. The similarity observed between the levels of the athletes might be a result of the different tests and protocols used by the researchers or might be evidence that the anthropometric and physiological profile of an MMA athlete did not vary among high-level athletes. Considering the increasing number of those engaging in MMA training and sport events, the findings of the present study provided strength and conditioning trainers working with MMA athletes a valuable tool for monitoring training and performance.

## 1. Introduction

Modern mixed martial arts (MMA) originated from the ancient Greek sport of Pankration, which was a combination of boxing and wrestling and was introduced for first time at the 33rd Olympic Games, in 648 BC [1]. Pankration’s name was a combination of the two Greek words, “pan”, meaning all, and “kratos”, meaning power [2]. MMA combined various martial arts relying on both striking and grappling, such as Brazilian jiu jitsu (BJJ), wrestling, boxing, kick-boxing, muay Thai, and many other styles [3]. MMA was first introduced by Gracie (BJJ expert) and Davie in 1993, by the organization known as the “Ultimate Fighting Championship” (UFC) [4]. The first competition organized was called “UFC 1”, had almost no rules, and was mainly a competition between different fighting styles, in which Gracie managed to prevail. Thereafter, the rules of the sport evolved for the protection of the athletes. MMA athletes were equipped with a mouthpiece, a groin or a chest protector for male or female athletes, respectively, and 4 oz gloves [5]. The UFC television events attracted more viewers than the National Basketball Association, National Hockey League, and Major League Baseball games, and their pay-per-view incomes were similar to major boxing and wrestling events [2].

The bout length of MMA ranged from three (regular bout) to five rounds (competition bout) consisting of five minutes each, with one minute of rest in between each round [5]. Since the duration of the bout was 15 to 25 min, the physiological demands derived mainly from the aerobic energy system [6]. In a single round of MMA, athletes could make multiple transitions from striking to grappling techniques; therefore, they should excel in various martial arts techniques. Also, short-term high-intensity explosive actions, such as landing strikes during ground fighting, have been shown to be decisive for the match outcome, highlighting the importance of anaerobic pathways [7,8]. 

MMA has been characterized by a combination of actions of high intensity and short duration such as striking and grappling. These actions highlighted the need for high levels of muscle power [9]—of both upper and lower limbs—and muscle strength—both dynamic [10] and isometric [9]—as well as high levels of muscular and aerobic endurance [11]. Given the multivariate nature of the sport and the fact that it was a “new” combat sport compared to the martial arts of which it was composed, many coaches and athletes designed their training programs in relation to the profile of other combat sports, whose anthropometric and physiological profiles scientific literature had been already established, such as wrestling [12], boxing [13], kick-boxing [14], and BJJ [15]. To this date, only two attempts have been made to gather the existing evidence and construct the anthropometric and physiological profile of an MMA athlete, and both attempts collected literature from the composing sports, and not the sport itself [16,17]. As Andreato and Branco pointed out, “such results cannot be extended to MMA specifically” [18]. Although the findings of the existing literature on other martial arts could not be transferred directly in MMA practice, they could be used as indicative of the optimal physiological characteristics in MMA. Such knowledge would be of great practical value for coaches and fitness trainers considering the popularity of this sport [19,20]. Being aware of the sport-specific physiological profile might help sport practitioners in terms of talent identification, athlete selection, and monitoring of training. Therefore, it was essential for sport practitioners to know the anthropometric and physiological profiles of MMA athletes with data that were derived directly from the sport, so they could properly design their training plans according to the specificity of the sport.

## 2. Materials and Methods

A computerized search was performed (January 2019) in the PubMed and Google Scholar databases for peer- reviewed investigations using the term “mixed martial arts” or “MMA” alone and in combination with “physiological characteristics”, “physiological profile”, “body composition”, “strength”, “power”, “endurance”, “aerobic”, and “anaerobic”. The abstracts of the relevant titles were then read, and if the study included anthropometric and/or physiological evaluation(s) for MMA athlete(s), the full text was retrieved. Manual searches were also done using reference lists from the recovered articles. 

## 3. Results

Nineteen articles matching our criteria were identified (Figure 1). Two relevant studies found were in Portuguese, one of them with an English abstract, and so personal contact was made with their authors wherever clarification was needed.

## 4. Discussion

### 4.1. Anthropometric Characteristics

The subjects of all the studies found were males (Table 1). The sample size ranged from 1, in two case studies [9,22], to 18 subjects [23]. The age range of the subjects was from 23 ± 4 years in a case study of UFC professional athletes [24] to 33 years [9]. It should be mentioned that the ranges reported hereafter referred to mean values, i.e., the abovementioned age 23 years was not of the youngest subject, but the youngest mean age. Height ranged from 170 [25] to 189 cm [9], and weight from 76.8 ± 9.4 to 90.2 kg [11]. Body mass index (BMI)—weight/height^2^—had values between 26 ± 3.3 [26] and 34.3 [22]. Body fat percentage (BF) of MMA athletes ranged from 8.5% [11] to 14.9 ± 7.2% [23]. Various methods for measuring BF were used, such as dual-energy X-ray absorptiometry (DEXA) [27,28], air displacement plethysmography [23], bioimpendance [22], and skin fold thickness [11,25,26,29,30], with the last being the most popular. It is worth emphasizing that the methods and the protocols used in order to calculate BF in MMA athletes were not the same; therefore, comparisons between athletes of different studies should be made with caution. The levels of BF found were similar to other martial arts, kick-boxing [14], boxing [13], and BJJ [15,31], but higher than wrestling [32]. In addition, it should be highlighted that low BF was combined with high BMI—i.e., increased weight for a given height—indicating that MMA athletes were characterized by large fat-free mass.

Based on the above-mentioned data, two observations could be made. First, there was a lack of research on female fighters. The only research found linking MMA with female athletes was the study of Schick and colleagues [33], which highlighted the strengths and conditioning considerations for the female athletes. The above-mentioned observation seemed to be a general trend among martial arts [12,13,14,31,32], perhaps because female athletes were not attracted to sports that include physical contact, aggressiveness, and face-to-face opposition [34]. Second, 16 out of the 17 studies presented had a mean sample age ≥25 years old, so it was apparent that there was also a lack of research regarding younger athletes. Perhaps this was because the UFC, the most prestigious MMA tournament up to date, was only open to adult participants. Additionally, most MMA fighters came from various disciplines of martial arts and were in the second stage of their career, thus, were older. Additionally, the perceived violence of the sport had contributed to it being banned in many states in the USA and in various countries around the world, even though the injury rate of MMA was similar to other combat sports [35]. This perception might prevent parents from enrolling their kids in MMA classes from an early age.

### 4.2. Muscular Strength

The strength of each subject was measured by various methods. The most popular among them was the one repetition maximal (1RM) bench press exercise [10,25,26,29] executed with a free barbell. Only one study used a Keiser pneumatic resistance strength machine [28] (Table 2). The 1RM squat exercise [10,26,29,30] was also used. It is worth noting that the authors used different depths in testing the 1RM squat exercise. Schick et al. [30] used a half-squat (knee angle = 90°), in contrast with Lachlan et al. [10] and Del Vecchio and Ferreira [25], who used a deep squat (knee angle > 90°). Marinho et al. [26,29] did not describe their exact protocol. As a result, these values must not be compared, because according to scientific literature, the load potential was greater in a half than in a deep squat exercise [36]. Thus, 1RM tests were used to evaluate muscle strength in both upper (bench press) and lower limbs (squat).

In addition to absolute (kg) values, 1RM lifts were presented also in relative values (kg/weight) in order to partition out the effect of body weight. The absolute values for the bench press, squat, and dead lift ranged from 76 ± 23 [29], 69 ± 6 [26], and 115 ± 10.69 [25] to 86 ± 17.847, 73 ± 15 [29], and 137 kg [37]. The relative values ranged from 0.93 ± 2.07 [29], 0.8 ± 0.1 [26], and 1.54 ± 0.27 [25] to 1.21 ± 0.18, 1.84 ± 0.23 [10], and 2.2 ± 0.19 [27], respectively. The values found in the above exercises were similar to the values found for wrestlers [12], boxers [13], and BJJ athletes [38]. Other exercises used for strength evaluation were the leg press and the seated row exercises [28], with values of 321.7 ± 41.8 and 80.5 ± 11.9 kg, respectively. 

The handgrip (HG) strength test [9,11,23,24,28,39,40] and the isometric middle-thigh pull (IMTP) [9,10,40] were used for evaluating the isometric strength of the fighters (Table 3). The HG isometric maximal strength for the dominant hand presented values from 45.8 ± 6.2 [30] to 78.4 kg [9]. The last reported value should be considered with caution, because the authors did not mention the protocol used for the execution of this test, and various parameters, such as body or hand position, can have an important effect on the results [41]. The HG values mentioned were similar to values found in BJJ [15], boxing [13], kick-boxing [14], and wrestling athletes [12]. The IMTP was measured via a custom-made apparatus [10], a dynamometer [40], and an AMTI lab force plate [9]. The relative values ranged from 25.87 ± 3.98 [10] to 37.7 N/kg [9]. Coswig et al. [40] reported the absolute value of this test as 173.8 ± 23.5 kg. These findings indicated a profile of high muscle strength for athletes in martial arts.

The existing studies emphasized the evaluation of upper limbs strength and handgrip performance. Although both have been shown to be important factors in grappling sports [12,42], the fact must not be neglected that punch force originated in the lower limbs, then was generated by trunk rotation and the upper limbs [17,43]. This observation has been shown to be an important difference between high- and low-level strikers [43]. Lower limb strength was also crucial to grappling performance [8]. Additionally, no study included core strength tests, with the exception of the deadlift exercise (which really evaluated the total body strength) although it was a major part of many performance aspects. Future MMA studies should consider including a core strength evaluation—e.g., isokinetic testing of trunk muscles [44]—in their battery tests.

### 4.3. Power

Multiple tests were used for the measurement of upper and lower limbs power (Table 4). SJ and CMJ with arm swing was used by the majority of studies for the evaluation of lower limb power, with ranging values from 29.5 ± 6.3 [39] and 48.5 ± 7.49 [24] to 57.2 [9] and 60.6 ± 5.5 cm [28], respectively. For the measurement of upper and lower limb power, the most commonly used test was the Wingate anaerobic test (WAnT), consisted of 30 s maximal exercise against braking force (depending on body mass) on a cycle ergometer (lower limbs) or an arm-cranking ergometer (upper limbs). The main indices of WAnT were peak power (PP, the power output recorded in the first 5 s), mean power (MP, the average power output during the 30 s period), and fatigue index (FI, the percentage decline of power out in the last 5 s compared to PP). Reported relative values for PP and MP of upper limbs ranged from 7.79 [22] and 5.8 [11], to 10.45 and 8.21 [9] W/kg, respectively. Also, the fatigue index (FI) was mentioned by the WAnT test in two studies, with values of 43.4% [9] and 58.1% [11]. 

In addition to the CMJ and the SJ, CMJ without arm swing was used with ranging values from 31.5 ± 6.2 [39] to 43.1 ± 5.07 cm [27]. Two studies by Marinho et al. also used the HJ to measure horizontal lower limb power with the results of 219 ± 25 [26] and 239 ± 31 cm [29]. Ghoul et al. [39] used a 10 m sprint to measure the same attribute, with a reported value of 2.03 ± 0.08 s. WAnT was used in one study [11] and reported the values of 10.2 W/kg MP, 7.6 W/kg PP, and 44.5% FI. These values are similar to wrestling values [12].

The hang power clean exercise, derived from Olympic weightlifting, was used by one study for total power body evaluation, with a relative value of 1.09 ± 0.07 kg/kg [27]. The BP throw with 60% of 1RM was used by Walker with a value of 601.8 W [37]. In a similar fashion, Coswing et al. [40] reported a value of 17.6 ± 4.9 cm in the plyometric pushup exercise. Barley et al. [24] mentioned a value of 474 ± 52 cm for an exercise in which a medicine ball (4 kg) was thrown from a seated position on the floor. In an effort to make a more sport-specific test using a laboratory environment, Bagley et al. [9] did five repetitions of the WAnT test, in order to mimic the five rounds of the fight, but unfortunately, we were not able to retrieve more details about the protocol used (such as time of effort, rest between the repetitions, etc.). Siqueido [28] also used a T-test, a multi-directional test that was commonly used for team sports, to evaluate the agility of the athletes, and reported a result of 10.3 ± 0.6 s. A sport-specific power evaluation measured three martial art strikes: The cross, the rear knee, and the double-leg strike had values of 5025.5, 2400, and 1852 W, respectively [37]. 

An interesting aspect of the specificity of the sport was the quality of “power-endurance” [17], which is defined as the application of a high degree of power, and a valid method to improving it is “by using relatively moderate to heavy loads (40–80% of 1RM) with the intention of moving the weight as quickly as possible” [46]. This quality described the relationship between exercise intensity (power) and the maximum period over which an exercise task can be sustained (endurance) at a constant power [47]. In contrast to measures of muscle endurance (presented in Section 4.4) that focused on the repeated exertion of submaximal force against a resistance, “power-endurance” emphasized the aspect of velocity requesting the exertion of submaximal force at maximal velocity [48]. Barley et al. [24] tested this quality with a repeated sled push test, in which a sled, loaded with 75% of the athlete’s weight, was pushed a distance of 10 m as fast as possible for 30 repetitions, with 20 s of active recovery between each push. The result reported was 29 ± 3 complete runs. An interesting version of the test could be the combination of a similar load in a total body movement effort reflecting the work-to-rest ratio of the sport [7].

A very interesting value was the FI % reported by WAnT of both upper and lower limb power, which has been highly correlated with anaerobic threshold [49], and as an influence of the performance [50]. Power was a key component for MMA athletes [51], and many times was a highly contributing factor to victory [8], and therefore must always be evaluated. However, only 1 out of 11 tests used in the studies above was sport-specific, including upper and lower limb strikes [37]. Additionally, two other studies tried to mimic sports physiological qualities [24] and competition round times [9]. Future studies should focus on finding sport-specific evaluations, which can directly test the sport-specific parameters. 

### 4.4. Muscular Endurance

The muscular endurance quality of MMA athletes was evaluated via simple and practical tests (Table 5). For assessing the isometric endurance of upper limbs, Marinho et al. [29] used the flexed arm chin-up hang, with results of 34 ± 11 and 35 ± 10 [18] s. The maximum number of pushups accomplished in 1 min were also used for testing upper limbs endurance, with results ranging from 37 ± 9 [26] to 41 ± 9 [29]. For evaluating the muscular endurance of the core, a test similar to the push-up test was used. The maximum number of sit-ups accomplished in 1 min ranged from 42 ± 14 [26] to 48.6 ± 5.3 [28]. The values reported for the pushup and the sit-up test were similar to BJJ values [31,38], but lower than those for wrestling [12]. Also, Coswig et al. evaluated endurance by the hang grip test [40], using 70% of the maximal isometric force and holding that for the maximum time, with a value of 37 ± 7.4 s for the dominant hand. One more test was used in the study of Alm [27]: The vertical sit-up test, which we assume has a similar protocol with the sit-up test, reported a value of 22.25 ± 3.78 repetitions. 

Once again, the lack of test specificity for muscular endurance in the sport of MMA was apparent, and future studies should aim to fill this gap. Sport-specific tests including strikes, throws, or other technical skills could be more accurate and helpful to coaches and may be more applicable to the athletes.

### 4.5. Cardiorespiratory Endurance

The cardiorespiratory fitness of MMA athletes was measured by maximal oxygen uptake (VO_2_max) tests, i.e., graded exercise tests on specialized ergometers either for upper (arm cranking) or lower limbs (Table 6). VO_2_max ranged from 44.2 ± 6.7 [23] to 62.8 ± 4.9 and from 45.8 ± 2.9 [27] to 55.0 mL/min/kg [11], for the lower and upper limbs, respectively. VO_2_max values mentioned for the lower limbs were higher than for BJJ [15] and for wrestling [12], and similar to muay Thai [52], boxing [13], and kick-boxing values [14]. The HRmax had values from 182 [22] to 193 bpm [11], results similar to those reported for muay Thai athletes [52].

In addition to VO_2_max, other measures of cardiorespiratory endurance were the absolute values of heart rate and VO_2_, and the % of maximal heart rate (HRmax) and VO_2_max at anaerobic threshold. De Oliveira et al. [23] reported HR 175.1 ± 11.5 bpm, VO_2_ 37.3 ± 6.8 mL/min/kg, and speed 13.6 ± 1.2 km/h at anaerobic threshold, which were similar to values reported for muay Thai athletes [52]. Alm [27] also mentioned the %HRmax and %VO_2_max at anaerobic threshold, which were 92 ± 2% of HRmax and 81 ± 9% of VO_2_max. Tota et al. [22] recorded values of 90% of HRmax and 74% of VO_2_max at anaerobic threshold. The levels of lactate concentration ([La]) 3 min after the end of exercise were also mentioned in two studies, reporting values of 14.6 ± 1.7 [27] and 10.0 [22] mmol/L. Lastly, the running speed at VO_2_max was reported in one study with a value of 15.4 ± 1.2 km/h [23].

Seven studies evaluated an athlete’s endurance based on VO_2_max values, upper and/or lower limbs, but only three studies reported values in relation to anaerobic threshold. Studies have shown that VO_2_max may not be the best qualification to consider for high-level athletes, and that anaerobic threshold value correlated better with performance [50,53]. Although, according to the duration of an MMA bout, the energetic demands derived mainly from the aerobic system, it should be noted that the outcome of the match was often decided by explosive actions, which were based on anaerobic pathways. As a result, future studies might rely their evaluation or anaerobic threshold markers and complement VO_2_max values. In summary, MMA athletes were characterized by cardiorespiratory endurance higher than the general population, but lower than endurance athletes [54].

### 4.6. Flexibility 

Flexibility referred to the ability of a muscle to lengthen allowing a joint to move through a range of motion. Two studies measured flexibility using the sit and reach test, and the values were 30.3 ± 10.6 cm [30] and 30.7 ± 7.3 cm [23] (Table 7). Siqueido [28] used the modified sit and reach test, which takes into account differences in limb length, and reported a value of 36.3 ± 8.2 cm. The values mentioned in the sit and reach test were similar to values reported for BJJ athletes [38] and wrestlers [12]. Another method was the adaptive flexitest [29]; the authors used eight of them in the current study, with a score of 18.38 ± 4.07. The adaptive flexitest was the sum of the score of various body movements [55]. The study by Siqueido [28] used a goniometer to measure shoulder flexion, with similar scores observed for the right and left shoulders. These findings suggested that MMA is characterized by high levels of flexibility [54]. It has been suggested that a high level of flexibility might result from the long-term engagement in MMA training [54].

### 4.7. Relationship of Anthropometric and Physiological Characteristics with Sport Performance

The analysis of the present study should be considered within the context of the relationship of anthropometric and physiological characteristics with sport performance. Upper and lower body anaerobic power and aerobic capacity have been observed to have small ability to predict performance in the Special Judo Fitness Test [56,57]. On the other hand, it should be highlighted that sport performance relied not only on the abovementioned characteristics, but also on other factors, such as technique and psychological characteristics [58,59]. In addition, the predicting ability of exercise testing variables might vary depending on which variables were modeled and the chosen martial arts [60]. Thus, although exercise testing variables reported in our review have been shown to properly evaluate the effectiveness of exercise training [61,62], the prediction of sport performance from anthropometric and physiological characteristics should be avoided. For instance, in a comparison of high- and low-intensity exercise programs, muscle mass in MMA was increased by high-intensity exercise [63]. Furthermore, the variation of the anthropometric and physiological characteristics by weight category should be considered when performing exercise testing. For instance, anaerobic power assessed by the WAnT in upper limbs was larger in absolute values (W) in heavier compared to lighter judo athletes, but there was no difference among weight categories once anaerobic power was expressed in relative to body mass values (W/kg) [64].

### 4.8. Limitations and Practical Applications

A limitation of the present review was that the sample size of the examined original research was relatively small, and the overall number of studies on MMA literature was also small; thus, caution is needed to make generalizations regarding the anthropometric and physiological profile for male athletes. Moreover, no studies included female MMA athletes; therefore, we have no evidence for what their anthropometric and physiological profiles might be. Furthermore, the majority of the above-mentioned studies used different tests, making it difficult to compare their results. The similarity observed between performance levels of the athletes might be a result of the different tests and protocols used by the researchers or might be evidence that the anthropometric and physiological profile of a martial arts athlete did not vary among high-level athletes. It would be assumed that other aspects of performance, like technique proficiency or various psychological indexes, would also be as important as the anthropometric and physiological characteristics. Considering the increasing number of those engaging in MMA training and sport events [18], the findings of the present study provided strength and conditioning trainers working with MMA athletes a valuable tool for monitoring training and performance. The assessment of anthropometric and physiological characteristics had practical applications for the effectiveness of training programs, talent identification, and selection of athletes. Thus, strength and conditioning trainers working with MMA athletes would be encouraged to use the normative data presented in this review considering the above-mentioned limitations.

## 5. Conclusions

The main findings of this review suggested that MMA athletes were characterized by average stature, increased body mass due to excess of fat-free mass, and low body fat. They had average cardiorespiratory endurance, but high levels of flexibility, muscle strength, muscle endurance, and anaerobic power. Although the relatively small sample size of participants in the original studies included in the present review did not allow meaningful comparison between different performance levels, the findings might be used to monitor effectively training.

## Figures and Tables

**Figure 1 sports-07-00146-f001:**
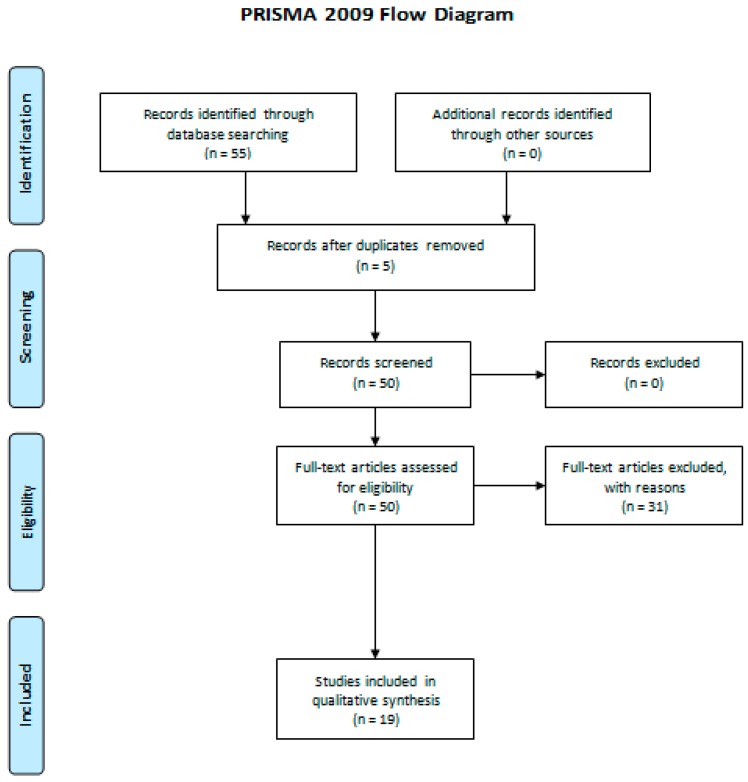
Prisma 2009 flow diagram [21].

**Table 1 sports-07-00146-t001:** Anthropometric characteristics of male mixed martial arts (MMA) athletes.

Participants and Level	Body Mass (kg)	Height (cm)	Body Fat (%)	Method	Details	Reference
National (N = 8)	82.1 ± 9.6	177 ± 5	13.4 ± 5.6	Skinfold thickness	triceps, abdomen, subscapular, medial calf	[26]
Regional (N = 8)	76.05 ± 10.27	170 ± 0.06	9.47 ± 4.06		triceps, suprailiac, thigh, pectoral, subscapular, abdominal	[25]
Professional (N = 1)	90.2	182	8.5		triceps, abdominal, subscapular, supraspinale, front thigh, medial calf	[11]
National (N = 13)	82.1 ± 10.9	176 ± 0.05	11.87 ± 5.11		abdomen, triceps, subscapular	[29]
Amateur (N = 11)	77.4 ± 11.4	174.8 ± 5.3	11.7 ± 4		triceps, chest, midaxilliary, subscapular, suprailliac, abdominal, thigh	[30]
Elite (N = 5)	80.8 ± 11.08	180.4 ± 9.07	12.25 ± 0.54	DEXA		[27]
Regional (N = 11)	80.3 ± 7.1	176.7 ± 6.8	12.3 ± 5.8			[28]
Elite (N = 1)	88.8	178	14.86	Bio-impedance		[22]
Regional (N = 18)	78.4 ± 6.95	172.5 ± 5.1	14.9 ± 7.2	Air displacement plethysmography Boyle		[23]

**Table 2 sports-07-00146-t002:** Dynamic strength characteristics of male MMA athletes.

Participants and Level	Body Mass (kg)	Strength Test	Absolute Value	Relative Value	Reference
			Kg	Mass/kg	
Semi-pro (N = 15)	79.8 ± 10.46	Bench press	N.R.	1.21 ± 0.18	[10]
Semi-pro and Amateurs (N = 14)	82.3 ± 12.5		N.R.	1.07 ± 0.2	
National (N = 13)	82.1 ± 10.9		76 ± 23	0.93 ± 2.07	[29]
National (N = 8)	82.1 ± 9.6		80 ± 15	1 ± 0.2	[26]
Regional (N = 11)	80.3 ± 7.1		86 ± 17.8	N.R.	[28]
Amateur (N = 11)	77.4 ± 11.4		N.R.	1.2 ± 0.1	[30]
Regional (N = 8)	76.05 ± 10.27		76.25 ± 10.61	1.01 ± 0.13	[25]
Intermediate and advance (N = 19)	N.A.		96	N.R.	[37]
Semi-pro (N = 15)	79.8 ± 10.46	Squat	N.R.	1.84 ± 0.23	[10]
Semi-pro and Amateurs (N = 14)	82.3 ± 12.5		N.R.	1.56 ± 0.24	
National (N = 13)	82.1 ± 10.9		73 ± 15	0.89 ± 1.38	[29]
National (N = 8)	82.1 ± 9.6		69 ± 6	0.8 ± 0.1	[26]
Amateur (N = 11)	77.4 ± 11.4		N.R.	1.4 ± 0.1	[30]
Elite (N = 5)	80.8 ± 11.08	Deadlift	N.R.	2.2 ± 0.19	[27]
Regional (N = 8)	76.05 ± 10.27		115 ± 10.69	1.54 ± 0.27	[25]
Intermediate and advance (N = 19)	N.A.		137	N.R.	[37]
Regional (N = 11)	80.3 ± 7.1	Leg press	321.7 ± 41.8	N.R.	[28]
		Seated row	80.5 ± 11.9	N.R.	

**Table 3 sports-07-00146-t003:** Isometric strength characteristics of male MMA athletes.

Participants and Level	Body Mass (kg)	Isometric Strength Test	Absolute Value	Relative Value	Reference
			Kg	Mass/kg	
Elite (N = 1)	102.1	Isometric HG strength (dominant hand)	78.4	N.R.	[9]
Amateur (N = 11)	77.4 ± 11.4		45.8 ± 6.2	N.R.	[30]
Regional (N = 12)	86.2 ± 10.9		52 ± 10.6	N.R.	[39]
Regional (N = 13)	81.3 ± 9.5		48.7 ± 7.1	N.R.	[40]
Regional (N = 11)	80.3 ± 7.1		52.1 ± 8.3	N.R.	[28]
Regional (N = 18)	78.4 ± 6.95		45.9 ± 8.9	N.R.	[23]
Amateur (N = 14)	76.8 ± 9.3		56 ± 10	N.R.	[24]
		IMTP	Kg	N/kg	
Elite (N = 1)	102.1		N.R.	37.7	[9]
Semi-pro (N = 15)	79.8 ± 10.46		N.R.	25.87 ± 3.98	[10]
Semi-pro and Amateurs (N = 14)	82.3 ± 12.5		N.R.	26.41 ± 5.34	
Regional (N = 13)	81.3 ± 9.5		173.8 ± 23.5	N.R.	[40]

**Table 4 sports-07-00146-t004:** Power characteristics of male MMA athletes.

Participants and Level	Test	Result	Reference
Elite (N = 1)	SJ (cm)	57.2	[9]
Regional (N = 12)		29.5 ± 6.3	[39]
Regional (N = 13)		43.8 ± 6.7 *	[40]
Elite (N = 5)		40.3 ± 3.78	[27]
Elite (N = 5)	CMJ (cm)	50.18 ± 5.63	[27]
Regional (N = 11)		60.6 ± 5.5	[28]
Amateur (N = 11)		57.6 ± 7.3	[30]
Amateur (N = 14)		48.5 ± 7.49	[24]
Intermediate and advance (N = 19)		59.69	[37]
Semi-pro (N = 15)		53.07 ± 8.48	[45]
Regional (N = 12)	CMJ without arm swing (cm)	31.5 ± 6.2	[39]
Elite (N = 5)		43.1 ± 5.07	[27]
National (N = 13)	HJ (cm)	219 ± 25	[7]
National (N = 8)		239 ± 31	[1]
Regional (N = 12)	10 m sprint (sec)	2.03 ± 0.08	[39]
	WAnT (Upper limbs)		
Elite (N = 1)	PP (W/kg)	10.45	[9]
Professional (N = 1)		8.9	[11]
Elite (N = 1)		7.79	[22]
Elite (N = 1)	MP (W/kg)	8.21	[9]
Professional (N = 1)		5.8	[11]
Elite (N = 1)		6.9	[22]
Elite (N = 1)	FI (%)	43.39	[9]
Professional (N = 1)		58.1	[18]
	Wingate test (Lower limbs)		
Professional (N = 1)	PP (W/kg)	10.2	[11]
	MP (W/kg)	7.6	
	FI (%)	44.5	
Elite (N = 5)	Hang Power Clean (mass/kg)	1.09 ± 0.07	[27]
Intermediate and advance (N = 19)	BP Throw (60% 1RM) (W)	601.8	[37]
Regional (N = 13)	Plyometric Push-up (cm)	17.6 ± 4.9	[40]
Amateur (N = 14)	Medicine Ball Throw (4 kg) (cm)	474 ± 52	[24]
Regional (N = 11)	T-test (sec)	10.3 ± 0.6	[28]
Amateur (N = 14)	Repeated 10 m Sled Push (complete runs)	29.29 ± 2.68	[24]
	Strikes (W)		
Intermediate and advance (N = 19)	Cross	5025.5	[37]
	Rear Knee	2400	
	Double Leg Strike	1852	

SJ = squat jump, CMJ = countermovement jump with arm-swing.

**Table 5 sports-07-00146-t005:** Muscular endurance characteristics of male MMA athletes.

Participants and Level	Body Mass (kg)	Test	Result	Reference
Regional (N = 11)	80.3 ± 7.1	Pushups (reps/min)	37.6 ± 6.8	[28]
National (N = 8)	82.1 ± 9.6		37 ± 9	[26]
National (N = 13)	82.1 ± 10.9		41 ± 9	[29]
Regional (N = 11)	80.3 ± 7.1	Sit-ups (reps/min)	48.6 ± 5.3	[28]
National (N = 8)	82.1 ± 9.6		42 ± 14	[26]
National (N = 13)	82.1 ± 10.9		43 ± 11	[29]
National (N = 8)	82.1 ± 9.6	Flexed Arm Chin-up Hang (sec)	35 ± 10	[26]
National (N = 13)	82.1 ± 10.9		34 ± 11	[29]
Regional (N = 13)	81.3 ± 9.5	Hang Grip Endurance (70%) (sec)	37 ± 7.4	[40]

**Table 6 sports-07-00146-t006:** Cardiorespiratory endurance characteristics of male MMA athletes.

Participants and Level	VO_2_max (mL/min/kg)	Details	Reference
Elite (N = 5)	62.8 ± 4.9 *	Standardized warm up. Initial speed 5 km/h. 7, 9, 10.6, 12 km/h every 3rd minute till exhausted. If not exhausted, then +2% inclination every 3rd minute.	[27]
Regional (N = 11)	55.4 ± 6.6	Bruce Protocol	[28]
Regional (N = 18)	44.2 ± 6.7	5′ warm up at 8 km/h. Initial speed of 9 km·h-1 with an increase of 1 km/h per 1′.	[23]
Regional (N = 8)	52.5 ± 55.0	2-3′ warm up at 6 km/h. Initial speed 8.5 km/h for 2′and then +1 km/h every 2′.	[25]
Professional (N = 1)	55.0	3′ warm up at 6 km/h. Initial speed 8 km/h. Then +1 km/h every 1′. After 16k m/h, +inclination until exhaustion.	[11]
Amateur (N = 11)	55.5 ± 7.3	5′ warm up at Monark Cycle, Gerkin Protocol	[30]
Elite (N = 1)	57.1	4′ warm up at 8 km/h with 1% inclination. Then an increase in speed by 1 km/h per 2′. When HRmax is reached, then stable speed and +1% inclination per 1′.	[22]
Professional (N = 1)	55.0 (Upper limbs)	2′ warm up at 45 W. Initial power at 60 W. Then +1w every 5″ until exhaustion or rpm < 60.	[11]
Elite (N = 5)	45.8 ± 2.9 (Upper limbs)	Initial power at 40 W. Then +30 W per 3 min until exhaustion.	[27]

* Values refer to a graded exercise test for lower limbs except if it is stated differently. rpm = revolutions per minute.

**Table 7 sports-07-00146-t007:** Flexibility of MMA athletes.

Participants and Level	Test	Result	Reference
Regional (N = 18)	Sit and reach (cm)	30.7 ± 7.3	[23]
Amateur * (N = 11)		30.3 ± 10.6	[33]
Regional (N = 11)	Modified sit and reach (cm)	36.3 ± 8.2	[28]
National (N = 13)	Adaptive flexitest (score)	18.38 ± 4.07	[29]
	Goniometer (shoulder flexion)	Degrees	
Regional (N = 11)	Right shoulder	169.2 ± 5.3	[28]
	Left shoulder	169.8 ± 4.6	

* Females.

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
