# Peer review of "Anthropometric and Physiological Profile of Mixed Martial Art Athletes: A Brief Review"

_sports, 2019, doi:10.3390/sports7060146_

Round 1
Reviewer 1 Report
Congratulation for your nice work
Author Response
Reviewer 1
Congratulation for your nice work
Answer: We thank the expert reviewer for the comment.
Reviewer 2 Report
Remarks
Some references should be add, as below.
On MMA physiology and training:
1. Chernozub A., Danylchenko S., Imas Y., Kochina M., Ieremenko N., Korobeynikov G., Korobeyjikova L., Potop V., Cynarski W.J., Gorashchenco A. (2019), Peculiarities of correcting load parameters in power training of mixed martial arts athletes, “Journal of Physical Education and Sport”, vol. 19, no. 2s, art. 70, pp. 481-488; doi: 10.7752/jpes.2019.s2070.
On profiles of other athletes – for discussion:
Mohammad Ali Boostani, Mohammad Hassan Boostani, Vali Nowzari (2012), Investigation and comparison of aggression in athletes in non-contact (swimming), limited contact (karate) and contact (kickboxing) sports, “Ido Movement for Culture. Journal of Martial Arts Anthropology”, vol. 12, no. 3, pp. 1–4.
I think, the psychological aspects and aggression of athletes should be analysed as a part of psychophysiological state; See:
1. Korobeynikov G., Cynarski W.J., Mytskan B., Dutchak M., Korobeynikova L., Nikonorov D., Borysova O., Korobeinikova I. (2019), The Psychophysiological State of Athletes with Different Levels of Aggression, “Ido Movement for Culture. Journal of Martial Arts Anthropology”, vol. 19, no. 1S, pp. 62–66; doi: 10.14589/ido.19.1S.10.
Author Response
Reviewer 2
Remarks
Some references should be add, as below.
On MMA physiology and training:
Chernozub A., Danylchenko S., Imas Y., Kochina M., Ieremenko N., Korobeynikov G., Korobeyjikova L., Potop V., Cynarski W.J., Gorashchenco A. (2019), Peculiarities of correcting load parameters in power training of mixed martial arts athletes, “Journal of Physical Education and Sport”, vol. 19, no. 2s, art. 70, pp. 481-488; doi: 10.7752/jpes.2019.s2070.
On profiles of other athletes – for discussion:
Mohammad Ali Boostani, Mohammad Hassan Boostani, Vali Nowzari (2012), Investigation and comparison of aggression in athletes in non-contact (swimming), limited contact (karate) and contact (kickboxing) sports, “Ido Movement for Culture. Journal of Martial Arts Anthropology”, vol. 12, no. 3, pp. 1–4.
I think, the psychological aspects and aggression of athletes should be analysed as a part of psychophysiological state; See:
Korobeynikov G., Cynarski W.J., Mytskan B., Dutchak M., Korobeynikova L., Nikonorov D., Borysova O., Korobeinikova I. (2019), The Psychophysiological State of Athletes with Different Levels of Aggression, “Ido Movement for Culture. Journal of Martial Arts Anthropology”, vol. 19, no. 1S, pp. 62–66; doi: 10.14589/ido.19.1S.10.
Answer: We agree with the expert reviewer and added the recommended references in the discussion. Please, see the requested changes in the text highlighted in blue.
Reviewer 3 Report
Comments:
- There are numerous errors of “Tense and Grammar” throughout the manuscript. Therefore, diligent editing is in order to fix the ENGLISH language.
- Insufficient literature review in the introduction.
- The authors investigated the review topic of mixed martial art athletes by applying highlights into the published articles for these sports, however, insufficient information used to applying the comparative overview between finding results of established studies. The material and methods presented with a weak explanation regarding the data collection. It could be better if the authors represent their vision again about the measurements or tools for assessing players in these sports.
- Line 93: It is not physical characteristics but anthropometric characteristics or anthropometric profile.
- Line 112: The authors must add to the title of table 1 the word male athletes because this table represents only data of males.
- Line 215: What is LT2?. Authors must define all abbreviations before
- Line 241 to 246: Unfortunately, the authors stated some wrong scientific expression about the cardiorespiratory profile. In this regard, WHAT is the meaning of VO2max tests for upper and lower limbs !!???
- In the discussion section, every table must continue a new column that refers to the sex type (Male or Female).
- Line 301: HOW authors can provide evidence of the validity of the talk in this line?
Author Response
Reviewer 3
Comments:
- There are numerous errors of “Tense and Grammar” throughout the manuscript. Therefore, diligent editing is in order to fix the ENGLISH language.
Answer: We agree with the expert reviewer and edited the text for English. Please, see the requested changes - for this one and the following comments - in the text highlighted in blue.
- Insufficient literature review in the introduction.
Answer: We agree with the expert reviewer and enhanced the introduction.
- The authors investigated the review topic of mixed martial art athletes by applying highlights into the published articles for these sports, however, insufficient information used to applying the comparative overview between finding results of established studies. The material and methods presented with a weak explanation regarding the data collection. It could be better if the authors represent their vision again about the measurements or tools for assessing players in these sports.
Answer: We agree with the expert reviewer and measurements and tools were further discussed.
- Line 93: It is not physical characteristics but anthropometric characteristics or anthropometric profile.
Answer: We agree with the expert reviewer and corrected it here and throughout the text.
- Line 112: The authors must add to the title of table 1 the word male athletes because this table represents only data of males.
Answer: We agree with the expert reviewer and corrected it here and in the next tables.
- Line 215: What is LT2?. Authors must define all abbreviations before
Answer: We agree with the expert reviewer and defined all the other abbreviations. For simplicity, we changed LT2 (lactate threshold 2) to anaerobic threshold here and throughout the text.
- Line 241 to 246: Unfortunately, the authors stated some wrong scientific expression about the cardiorespiratory profile. In this regard, WHAT is the meaning of VO2max tests for upper and lower limbs !!???
Answer: We agree with the expert reviewer and clarified this part (“The cardiorespiratory fitness of MMA athletes was measured by maximal oxygen uptake (VO2max) tests, i.e. graded exercise tests on specialized ergometers either for upper (arm cranking) or lower limbs”).
- In the discussion section, every table must continue a new column that refers to the sex type (Male or Female).
Answer: We agree with the expert reviewer and added this information in the titles of tables.
- Line 301: HOW authors can provide evidence of the validity of the talk in this line?
Answer: We agree with the expert reviewer and revised this part.
Reviewer 4 Report
Comments:
The review is relevant for elaborate training plans in this sport. The aim is clear, simple and is well justified in introduction. Authors took into account many methodological points (1RM normalized with fighter weight, tools to measure body fat mass etc…) to clarify the relevance of their results. Nevertheless, I have several comments:
Introduction
- The following sentence is hard to read. Please clarify it:
“The combination of striking and grappling requires the MMA athletes to be able to blend high levels of power [9] – _of both upper and lower limbs – _and strength – _both dynamic [10] and isometric [9] – with high levels of muscular and aerobic endurance [11].”
- Authors should determine the need to know the physical and physiological profile of MMA athletes in performance for coaches in order to help them to improve their training plans as mentioned at the end of the Introduction. It could reinforce the relevance of the review. Authors made the link between physical and physiological characteristics at the end of the manuscript and postulated that it is difficult to determine the role of these characteristics. It might be indicate in the introduction to be clearer. Therefore, what is the relevance of this review?
- Maximal weight was 90 kg. So, it might also consider as a limit of the review because the physical and physiological profile does not include the heavy weight athletes, who should likely have some different characteristics in terms of body fat mass or muscle weight for instance. It should be discussed.
Core of the manuscript
- For power analysis, the WAnT needs to be defined and briefly described. It is less common than squat jump or counter movement jump.
- Please clarify the definition of the quality of “power-endurance”: “which is defined as the application of a high degree of power, and a valid method to improving it is “by using relatively moderate to heavy loads (40-80% of 1RM) with the intention of moving the weight as quickly as possible”. Authors did not refer to the capacity to maintain a high degree of power during a given period (few minutes max). Is it different for the authors? It is similar to “force-endurance” or muscular endurance (as in the next section)? It is not clear in my opinion.
- Authors did not compare the different results of MMA athletes with other populations for VO2max, VT2, muscular endurance and other parameters. It is thus less relevant because readers have no idea if it is better or not for these athletes compared to other sports. It could help to highlight the specificity of these athletes.
Author Response
Reviewer 4
Comments:
The review is relevant for elaborate training plans in this sport. The aim is clear, simple and is well justified in introduction. Authors took into account many methodological points (1RM normalized with fighter weight, tools to measure body fat mass etc…) to clarify the relevance of their results. Nevertheless, I have several comments:
Introduction
- The following sentence is hard to read. Please clarify it:
“The combination of striking and grappling requires the MMA athletes to be able to blend high levels of power [9] – _of both upper and lower limbs – _and strength – _both dynamic [10] and isometric [9] – with high levels of muscular and aerobic endurance [11].”
Answer: We agree with the expert reviewer and clarified it. Particularly, we changed it to “MMA has been characterized by a combination of actions of high-intensity and short duration such as striking and grappling. These actions highlighted the need for high levels of muscle power [9] – of both upper and lower limbs – and muscle strength – both dynamic [10] and isometric [9] – as well as high levels of muscular and aerobic endurance [11].”. Please, see the requested changes - for this one and the following comments - in the text highlighted in blue.
- Authors should determine the need to know the physical and physiological profile of MMA athletes in performance for coaches in order to help them to improve their training plans as mentioned at the end of the Introduction. It could reinforce the relevance of the review. Authors made the link between physical and physiological characteristics at the end of the manuscript and postulated that it is difficult to determine the role of these characteristics. It might be indicate in the introduction to be clearer. Therefore, what is the relevance of this review?
Answer: We agree with the expert reviewer and addressed this aspect (rationale for the study) in the end of introduction before aim s (“Such knowledge would be of great practical value for coaches and fitness trainers considering the popularity of this sport [19,20]. Being aware of the sport-specific physiological profile might help sport practitioners in terms of talent identification, athletes’ selection and monitoring of training.”).
- Maximal weight was 90 kg. So, it might also consider as a limit of the review because the physical and physiological profile does not include the heavy weight athletes, who should likely have some different characteristics in terms of body fat mass or muscle weight for instance. It should be discussed.
Answer: We agree with the expert reviewer that this issue needs clarification. Actually 90kg refers to a case study, whereas other studies report mean ~80kg and SD ~10 indicating that they include heavier than 90kg athletes too. We clarified it reporting “It should be mentioned that the ranges reported hereafter referred to mean values, i.e. the abovementioned age 23 years was not of the youngest subject, but the youngest mean age.”.
Core of the manuscript
- For power analysis, the WAnT needs to be defined and briefly described. It is less common than squat jump or counter movement jump.
Answer: We agree with the expert reviewer and explained WAnT briefly (“Wingate anaerobic test (WAnT), consisted of 30 s maximal exercise against braking force (depending on body mass) on a cycle ergometer (lower limbs) or an arm-cranking ergometer (upper limbs). The main indices of WAnT were peak power (PP, the power output recorded in the first 5 s), mean power (MP, the average power output during the 30 s period) and fatigue index (FI, the percentage decline of power out in the last 5 s compared to PP).”).
- Please clarify the definition of the quality of “power-endurance”: “which is defined as the application of a high degree of power, and a valid method to improving it is “by using relatively moderate to heavy loads (40-80% of 1RM) with the intention of moving the weight as quickly as possible”. Authors did not refer to the capacity to maintain a high degree of power during a given period (few minutes max). Is it different for the authors? It is similar to “force-endurance” or muscular endurance (as in the next section)? It is not clear in my opinion.
Answer: We agree with the expert reviewer and clarified it (“This quality described the relationship between exercise intensity (power) and the maximum period over which an exercise task can be sustained (endurance) at a constant power [48]. In contrast to measures of muscle endurance (presented in the section 4.4) that focused on the repeated exertion of submaximal force against a resistance, “power-endurance” emphasized the aspect of velocity requesting the exertion of submaximal force at maximal velocity [49].”).
- Authors did not compare the different results of MMA athletes with other populations for VO2max, VT2, muscular endurance and other parameters. It is thus less relevant because readers have no idea if it is better or not for these athletes compared to other sports. It could help to highlight the specificity of these athletes.
Answer: We agree with the expert reviewer and added these comparisons.
Round 2
Reviewer 2 Report
It looks better now.
Reviewer 3 Report
Thank you for your understanding regarding my last review comments.
This manuscript is a resubmission of an earlier submission. The following is a list of the peer review reports and author responses from that submission.
Round 1
Reviewer 1 Report
This is a really nice research.
Congratulation to the authors
Reviewer 2 Report
Thank you for the opportunity to have appraised the submission titled: “Physical and physiological profile of mixed martial art athletes: A brief review” for Sports. The authors have attempted to review the literature surrounding the underpinning characteristics of MMA athletes. While the submission is mostly well written, I feel that there are a number of major issues which preclude acceptance. I summarise my comments as follows…
The abstract (and indeed the paper itself) omits mention of any real mention of the findings from the review; for example, only limitations appear presented in lines 17-22. Accordingly, the conclusions lack context.
While acknowledging the limitations of limited literature, the introduction does not develop a strong rationale for the study and seems more predicated on the absence of literature being the driver behind the paper as opposed to outlining the need to report the physical and physiological characteristics of this population of athletes. For example, are there any correlates of success or key performance indicators that are crucial for MMA (or if not, can be inferred from other sports)?
The primary concern given the structure of the paper is the absence of information relating to the methods and results sections. While I understand that this is not a systematic review, the approach taken in lines 68-74 suggest otherwise yet still omit pertinent information. For example, between what dates did the searches take place? What specific criteria were used in the selection of articles? Akin to the reporting of results, although 19 papers were retrieved, how many were discounted (and why)? Such information is important to replicate the search and selection of study findings.
In the majority of places the discussion section is very descriptive (e.g., lines 81-92) and like the abstract, seems to focus on the limitations of the literature base (with a view to recommending future research recommendations) before actual critique of the papers sourced. This is also representative of the conclusions of the paper which presently lack focus and from the outset seems to focus on the shortfalls of previous research.
Inconsistencies also exist in the presentation of data within single sentences and table formatting (e.g., table 6) is problematic throughout and hinders readability.
Reviewer 3 Report
Introduction:
LN 39: The term "Nowadays" is too colloquial.
LN 41-2: Change "are attracting" to attract.
LN 48: Need to provide examples of what "explosive actions" are. Further, move this sentence down as it seems out of place with how the following sentences are structured.
Materials and Methods/Results:
Significantly more detail about the systematic nature of the search is necessary. Adding a PRISMA Flow Chart would be a good first step at fulfilling the description of the systematic search for relevant articles. Further, it's unclear why Portuguese language articles were used when English Language was a search criteria for the articles. Suggest removing these articles from the results.
Discussion:
Overall, the description of the physical and physiological descriptors of the MMA athletes from the reviewed articles seems more appropriate in a results section than the Discussion. The discussion should be more of an interpretation of these results and how they may or may not be improved upon.
LN 82: The sentence that starts on this line is difficult to understand. Is the age range specific to the case studies (2), if so, why is there a different citation # than the 2 provided for the case studies in the preceding sentence?
LN110: Need to define what 1RM is before abbreviating it. This goes for all abbreviations throughout the manuscript. There are a number of abbreviations throughout the Discussion that appear to have no definition.
LN 140: Replace body with limb, for "Lower body strength...". There are a number of similar instances that require the word limb following upper/lower.
LN 143: Can the authors provide a suggestion for appropriate core strength assessment methodologies?
LN 245: MMA athletes are above average, compared to whom?
Conclusion:
LN 260: What levels are being referred to?
LN264: Add physical and physiological to the sentence to state that the physical and physiological characteristic of female MMA athletes remains unknown.
LN @76: The authors appropriately state that the results described within are limited to a small sub population of MMA fighters, specifically males, over the age of 25. Yet the concluding statement of the review generalizes the results to MMA athletes, which is a stretch.